# Effectiveness of accelerated diagnostic protocols for reducing emergency department length of stay in patients presenting with chest pain: A systematic review and meta-analysis

Jesse Hill[1,2], Nana Owusu M. Essel[2], Esther H. Yang[2,3], Liz Dennett[4], Brian H. Rowe[2,5]*

1 Department of Emergency Medicine, Misericordia Community Hospital, Edmonton, Alberta, Canada,
2 Department of Emergency Medicine, College of Health Sciences, University of Alberta, Edmonton, Alberta, Canada, 3 The Alberta Strategy for Patient-Oriented Research Support Unit, Alberta Health Services (AHS), Edmonton, Alberta, Canada, 4 Geoffrey and Robyn Sperber Health Sciences Library, College of Health Sciences, University of Alberta Edmonton, Edmonton, Alberta, Canada, 5 Department of Clinical Epidemiology, School of Public Health, College of Health Sciences, University of Alberta, Edmonton, Alberta, Canada

* browe@ualberta.ca

**Data Availability Statement:** All relevant data are within the manuscript and its Supporting Information files.

## Abstract

In recent years, there has been an increase in the use of accelerated diagnostic protocols (ADPs) and high-sensitivity troponin assays (hsTn) for the assessment of chest pain in emergency departments (EDs). This study aimed to quantitatively summarize the operational and clinical outcomes of ADPs implemented for patients with suspected cardiac chest pain. To be considered eligible for inclusion, studies must have implemented some form of ADP within the ED for evaluating adult (age ≥18 years) patients presenting with chest pain using Tn assays. The primary outcome was ED length of stay (LOS). Secondary outcomes included the proportion of patients admitted and the proportion with 30-day major adverse cardiac events (MACE). Thirty-seven articles involving 404,566 patients met the inclusion criteria, including five randomized controlled trials (RCTs) and 32 observational studies. A significant reduction in total ED LOS was reported in 22 observational studies and four RCTs. Emergency departments with longer baseline ED LOS showed significantly larger reductions in LOS after ADP implementation. This observed association persisted after adjusting for both the change in serial Tn measurement interval and transition from conventional Tn assay to an hsTn assay (β = -0.26; 95% CI, -0.43 to -0.10). Three studies reported an increase in the proportion of patients admitted after introducing an ADP, one of which was significant while 15 studies reported a significant decrease in admission proportion. There was moderate heterogeneity among the 13 studies that reported MACE proportions, with a non-significant pooled risk ratio of 0.95 (95% CI, 0.86−1.04). Implementation of ADPs for chest pain presentations decreases ED LOS, most noticeably within sites with a high baseline LOS; this decreased LOS is seen even in the absence of any change in troponin assay type. The decrease in LOS occurred alongside reductions in hospital admissions,

**Funding:** Dr. Jesse Hill has no funding sources to report. Dr. Brian Rowe's research is supported by a Scientific Director's Grant (SOP 168483) from the Canadian Institutes of Health Research (CIHR, Ottawa, Ontario). Ms. Esther Yang is supported by the Emergency Medicine Research Group (EMeRG) in the Department of Emergency Medicine, University of Alberta. There was no additional external funding received for this study. The funders take no responsibility for the conduct, analyses, and interpretation of these results.

**Competing interests:** The authors have declared that no competing interests exist.

while not increasing MACE. The observed benefits translated across multiple countries and health regions.

## Introduction

Emergency departments (EDs) are faced with one of the greatest challenges in modern healthcare systems: ED crowding [1], leading to ED staff being unable to provide timely and evidence-based care to ED patients [2]. Crowding in the ED has real consequences, both obvious and unintended. Increased patient wait times are perhaps the most readily apparent consequence of ED crowding and are often reported by the media. Delays in seeing a care provider, however, also lead to delays in time-sensitive treatments and poor patient outcomes [3] including increased mortality [4].

Chest pain is a high-volume ED presentation, the second most common in Canada [5], and has been a focus of interventions aimed at improving efficiency. Scoring systems for the assessment of chest pain have been developed to provide pre-test probabilities of serious outcomes [6]. In addition, protocols for assessing ED patients with chest pain have been implemented. Finally, rapid advances in cardiac biomarkers, such as troponin (Tn), and their availability in the ED have increased the confidence of ED physicians in using these protocols.

Tn accumulates in blood after cardiac muscle necrosis and rising levels are considered a surrogate marker of acute coronary syndromes (ACS). Early conventional Tn assays historically required at least 6-hour serial measurements to achieve adequate sensitivity as a rule-out tool for ACS. Given that the remainder of the ED workup for chest pain may take approximately 1–2 hours, the 6-hour serial Tn contributes to a longer ED length of stay (LOS). Over the past two decades, there have been advances in cardiac biomarker assays. Tn detection thresholds improved from 0.10 μg/L to 0.04 μg/L with the introduction of high-sensitivity Tn (hsTn) in 2007 [7]. This lower detection threshold was added to clinical decision rules to create accelerated diagnostic protocols (ADPs), with lower serial Tn measurement intervals. Ongoing improvements to hsTn measurements have led to detection thresholds as low as 2–3 ng/L, allowing rising levels to be detected with as little as a 1-hour serial measurement [8].

Given the intuitive possibility that using an ADP would increase ED efficiency, there has been an increase in studies pertaining to operational outcomes. Many studies have focused on the safety of ADPs, specifically their ability to determine which patients are at high risk of developing a major adverse cardiac event (MACE) within 30 days. Once the safety profile was determined to be acceptable, the focus in recent years has shifted to efficiency. The purpose of this study was to quantitatively summarize the operational and clinical outcomes (specifically ED LOS, admission proportions, and MACE) of ADPs implemented for patients with suspected cardiac chest pain.

## Methods

### Protocol

A protocol was developed *a priori* to define the objectives, search strategy, eligibility criteria, outcomes, procedure for extracting and analyzing information from included studies, and data analysis. This systematic review conforms to the Preferred Reporting Items for Systematic Reviews and Meta-Analyses (PRISMA) guidelines and was registered in PROSPERO (registration number: CRD42021249679).

## Search strategy

Bibliographic databases were comprehensively searched, each from inception to October 16, 2023, in accordance with the Meta-Analysis of Observational Studies in Epidemiology (MOOSE) guidelines, including Embase (OVID), LILACS, CINAHL (EBSCO), Scopus, MEDLINE (OVID), Theses Global (ProQuest), and Cochrane Trials (Wiley). The search strategy was conducted by an experienced health science librarian (LD) based on subject headings and keywords and optimized for each database; the full search strategy is available in the S1 File. No records were excluded based on the date of publication or language. Grey literature searches were conducted using Google Scholar, hand searches of conference abstracts, and search of clinical trial registries. Bibliographies of retrieved articles and known reviews were also searched for relevant studies.

## Study selection

To be considered eligible for inclusion, studies must have implemented some form of ADP within the ED for evaluating adult (age ≥18 years) patients presenting with chest pain. Common risk stratification tools used as part of ADPs included but were not limited to the History, Electrocardiogram, Age, Risk factors, and Troponin (HEART) pathway; Emergency Department Assessment of Chest Pain Score (EDACS); and Accelerated Diagnostic protocol to Assess chest Pain using Troponin (ADAPT). The primary outcome was ED LOS, defined as the time from triage to discharge (for discharged patients) or to bed request (for admitted patients). Secondary outcomes included the proportion of patients requiring admission and the proportion of patients with MACE, defined as a composite of total deaths, myocardial infarction (MI), stroke, and revascularization within 30 days of the ED presentation. Studies were required to be either randomized controlled trials (RCTs), controlled clinical trials (CCTs), before-after studies, or observational studies (prospective and retrospective) with a well-matched comparison group.

Two independent reviewers (EHY and NOME) identified relevant studies in a two-step process. First, from the title, abstract, or descriptors, we independently reviewed articles to identify potentially relevant studies for a full-text review. Second, from the full text, using specific criteria, we independently selected studies for inclusion in this review. Standardized forms with pre-defined inclusion/exclusion criteria were used. Disagreements were resolved by a third reviewer (JH); reasons for exclusion were documented.

## Risk of bias assessment

The risk of bias of RCT/CCTs was assessed using the Cochrane Risk of Bias (RoB) tool [9], which uses a fixed set of domains of bias that focus on different aspects of trial design, conduct, and reporting. Studies could be judged as having a "low" or "high" risk of bias or the reviewer could express "some concerns". The quality of observational cohort studies was assessed using the Newcastle−Ottawa assessment scale (NOS) [10], which uses a star system to appraise bias in three domains: participant selection, comparability among groups, and assessment of exposure/outcomes. Studies are rated from 0–9, which can be classified as high (NOS scores <4), intermediate (NOS scores 4–6), or low (NOS scores of ≥7) risk of bias, respectively. Pre-specified criteria were used to assign a risk of bias score to the included studies. Two reviewers (EHY and NOME) independently evaluated the methodological quality of the studies and disagreements were discussed and resolved with a third-party mediator (JH).

## Data extraction and statistical analyses

Data from the included studies were extracted independently by two reviewers (EHY and NOME). Disagreements and reliability were checked by a third reviewer (JH). Agreement was

measured and reported using Cohen's $\kappa$ statistic. Findings from these articles were tabulated, including information about each article's source, country of origin, year of publication, design, demographics, type of Tn assay used, ADP used, ED LOS, the proportion of patients admitted, and proportions of patients who experienced MACE. Where relevant data were missing from published articles, attempts were made to contact the authors to request further information.

Studies were pooled if they represented similar populations, outcomes, and designs, and were judged to have sufficiently low clinical heterogeneity. ED LOS as a continuous indicator was often skewed and thus reported as medians with interquartile ranges (IQRs) in a majority of the studies. Therefore, a meta-analysis was conducted with differences in median ED LOS by pooling differences of medians using a random-effects model via the quantile estimation (QE) method [11]. This method has been found to outperform transformation-based methods in meta-analyses of median outcomes, particularly when outcome measures are skewed [12]. In the secondary analyses, median ED LOS values in each subgroup were pooled via the same method. The QE methodology derives the variance for medians or differences of medians for studies reporting medians, first quartiles, and third quartiles for an outcome (scenario $S_2$) [13]. For three out of 37 studies that reported mean ED LOS values, the QE algorithm (scenario $S_4$) was applied to obtain differences in medians and their variances [13]. Between-study heterogeneity was assessed in each analysis using $I^2$ statistics with values of 25%, 50%, and 75% representing low, moderate, and high heterogeneity, respectively. In addition, a linear meta-regression analysis was performed to evaluate the effect of pre-interventional (baseline) ED LOS on the resulting change in ED LOS after ADP implementation. The estimate from this meta-regression was further adjusted for the delta (pre-post) serial Tn measurement interval and whether the ADP implementation involved a transition from conventional Tn to hsTn use. The 24 studies that reported admission proportions and thirteen studies that reported MACE proportions were pooled using the random-effects DerSimonian–Laird estimator with risk ratios (RRs) as measures of effect size.

For the meta-analysis of the primary outcome (ED LOS), we used the 'hot deck' method in the R package *metagear* to impute three (8%) missing standard deviation (SD) values for studies which failed to report this measure of spread. No imputation was pursued for the meta-analyses involving the secondary outcomes (MACE and admission proportion); all studies with missing data were excluded in those meta-analyses.

The statistical computations were performed using RStudio 2022.02.1 in the R 4.2.0 environment (The R Development Core Team, Vienna, Austria) and Stata version 18 (StataCorp LLC, College Station, TX, USA). The data from the analyses are presented descriptively if data pooling was prohibited due to high heterogeneity among the studies or insufficient reporting of outcomes. Readers will be alerted if there was substantial heterogeneity ($I^2 > 50\%$) and urged to interpret all aggregated results cautiously. Assuming sufficient heterogeneity and outcome reporting allowed for a meta-analysis, several sensitivity analyses were completed including fixed effects as well as study quality assessment (in which studies with a high risk of bias would be excluded).

## Results

### Search and study selection

Our search strategy generated 13,612 citations. After accounting for duplicates and performing a title and abstract screen, a total of 215 articles underwent full-text review. Thirty-seven articles [14–50] involving a total of 404,566 patients met the inclusion criteria ($\kappa = 0.86$; 95% CI: 0.71–0.95) (Fig 1). Of the 178 articles excluded at full-text review, the most common reasons

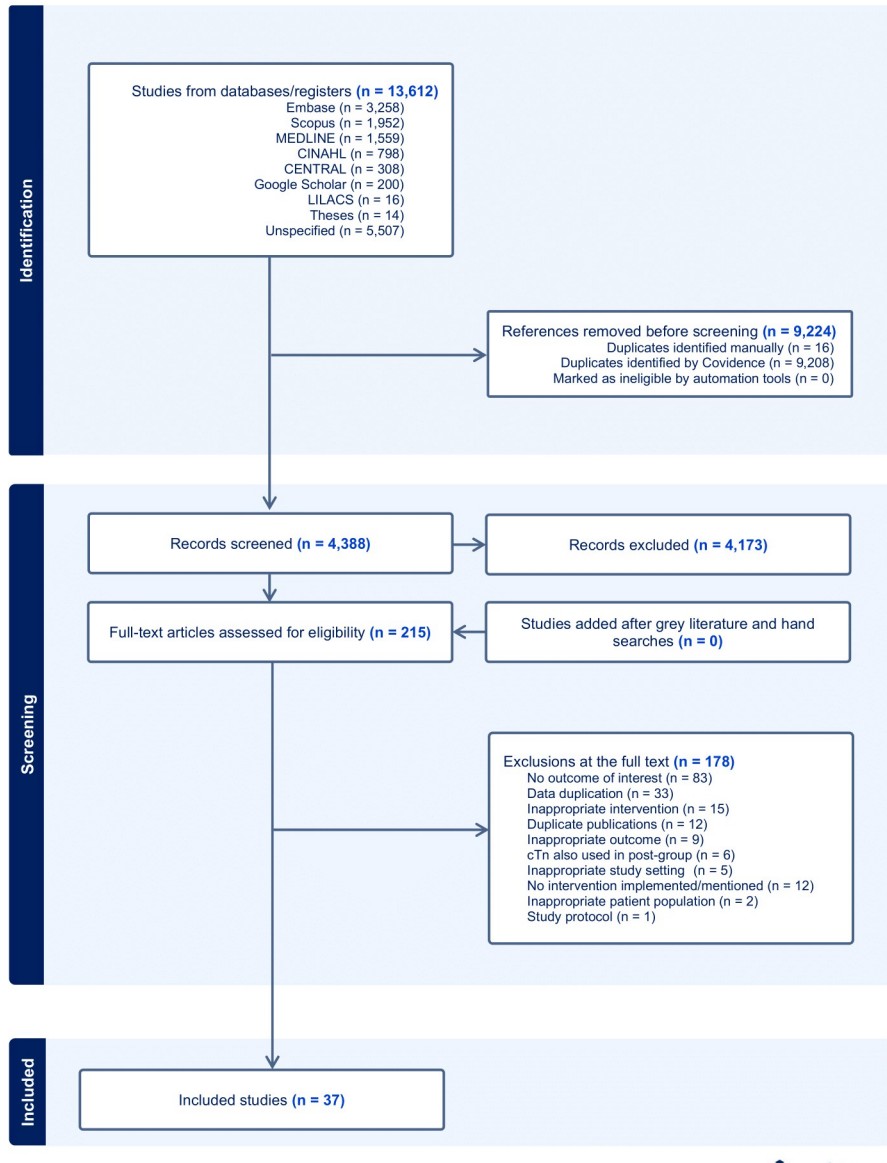

**Fig 1. PRISMA flow diagram illustrating the overview of the systematic literature search.**

were no outcome of interest (n = 83), data duplication in other studies (n = 33), inappropriate intervention (n = 15), and duplicate publications (n = 12) (Fig 1).

## Study characteristics

Table 1 provides a summary of the study characteristics. Briefly, the included studies represented a diverse patient group, drawn from populations on four different continents (North America, Asia, Europe, and Australia). Among the 37 included studies, there were five RCTs and 32 observational studies were identified. The characteristics of EDs also varied, ranging from sites with an annual census of 12,000 to 331,000 patients. The included studies were most frequently conducted in the United States of America (n = 18) and Australia (n = 6). Due to

**Table 1. Study design and demographic characteristics of the included studies.**

| Author | Publication year | Country | Sample size | Age: mean (±SD) or median (IQR) | | % females | |
|---|---|---|---|---|---|---|---|
| | | | | Pre | Post | Pre | Post |
| **Before-after studies** | | | | | | | |
| Al Marashi | 2020 | Australia | 634 | 61.3±13.6 | 58.7±12.0 | 48.0 | 53.0 |
| Allen | 2018 | USA | 31090 | 51.9±1.2 | 52.6±0.5 | 54.8 | 55.9 |
| Barnes | 2021 | Australia | 2255 | 55±17.0 | 52±17.0 | 47.0 | 47.0 |
| Bevins | 2022 | USA | 7844 | 59±16.0 | 59±15.0 | 46.0 | 45.0 |
| Buttinger | 2019 | USA | 882 | 61.0 | 57.0 | 41.0 | 42.0 |
| Crowder | 2015 | Canada | 12620 | 62.7 | 61.4 | 48.2 | 50.1 |
| Ford | 2021 | USA | 3205 | 54.0 (39.0–65.0) | 55.0 (41.0–66.0) | 50.0 | 49.0 |
| Furmaga | 2021 | USA | 12345 | 58.8±17.7 | 59.1±17.4 | 55.7 | 58.0 |
| Ganguli | 2021 | USA | 7564 | 56.3±17.1 | 55.7±17.4 | 53.0 | 54.7 |
| Greenslade | 2020 | Australia | 12630 | 61.0±17.2 | 58.1±17.0 | 41.6 | 41.2 |
| Hill | 2023 | Canada | 2640 | 58.0 (44.0–70.0) | 56.0 (43.0–68.0) | 47.5 | 42.8 |
| Hughes | 2023 | USA | 59232 | 60.0 (49.0–71.0) | 60.0 (50.0–70.0) | 49.5 | 49.4 |
| Ljung | 2019 | Sweden | 1233 | 64.0 (54.0–74.0) | 63.0 (53.0–71.0) | 43.0 | 46.0 |
| Mahler | 2018 | USA | 8474 | 54.0 (45.0–65.0) | 54.0 (44.0–66.0) | 52.9 | 54.1 |
| Mohmed | 2021 | UK | 3018 | 58.0 | 55.0 | 47.7 | 47.9 |
| Mountain | 2016 | Australia | 1029 | 64.0 | 64.0 | 50.3 | 52.4 |
| Mumma | 2020 | USA | 1078 | 60.0 (48.0–70.0) | 61.0 (51.0–71.0) | 52.4 | 47.8 |
| Mungai | 2020 | USA | 300 | 58.6±10.8 | 57.3±8.7 | 49.3 | 42.0 |
| Ola | 2021 | USA | 3536 | 62.0±18.0 | 61.0±18.0 | 51.0 | 52.0 |
| Parsonage | 2017 | Australia | 54468 | 60.6±16.0 | 58.9±16.5 | 45.8 | 46.0 |
| Phillips | 2023 | Canada | 11703 | 58.0 (44.0–72.0) | 57.0 (42.0–71.0) | 51.1 | 50.4 |
| Randolph | 2018 | USA | 5064 | - | - | 49.0 | 46.0 |
| Rowe | 2023 | Canada | 4339 | - | - | - | - |
| Ruangsomboon | 2018 | Thailand | 130 | 71.6±13.2 | 66.6±14.2 | 43.1 | 60.0 |
| Suh | 2022 | USA | 1892 | 60.3±15.8 | 60.4±15.9 | 50.2 | 54.4 |
| Than | 2018 | New Zealand | 31332 | 65.1±16.4 | 65.8±16.1 | 46.5 | 45.6 |
| Than | 2021 | New Zealand | 2416 | 63.0±13.0 | - | 38.2 | - |
| Trent | 2022 | USA | 1298 | - | - | - | - |
| Twerenbold | 2016 | Switzerland | 2544 | 64.0 (51.0–76.0) | 59.0 (47.0–72.0) | 32.0 | 30.0 |
| Tyner | 2023 | USA | 14740 | 49.6±14.6 | 49.3±15.3 | 56.0 | 55.2 |
| VanAssche | 2023 | Belgium | 200 | 63.0±18.0 | 59.0±19.0 | 34.0 | 46.0 |
| Vigen | 2020 | USA | 31543 | 53.8±14.2 | 54.2±14.6 | 48.1 | 47.0 |
| **RCTs** | | | | **Control** | **Exp.** | **Control** | **Exp.** |
| Anand | 2021 | Scotland | 31492 | 59.0±17.0 | 58.0±17.0 | 45.0 | 46.0 |
| Carlton | 2020 | UK | 629 | 53.6±16.2 | 54.0±16.2 | 41.0 | 41.0 |
| Chew | 2019 | Australia | 3288 | 58.6 (49.0–71.0) | 58.7 (49.0–69.0) | 46.8 | 46.8 |
| Lambrakis | 2021 | USA | 3270 | 58.6 (49.0–71.0) | 58.7 (49.0–69.0) | 46.7 | 46.8 |
| Miller | 2022 | USA | 32609 | - | - | - | - |

Exp., experimental group; SD, standard deviation; IQR, interquartile range; RCT, randomized controlled trial; Pre, before ADP implementation; Post, after ADP implementation.

the recency with which ADPs have entered clinical practice, there was a preponderance of recent publications, with 68% (n = 25) of included studies published after 2019. The sample sizes for individual studies ranged from 130 to 59,232 with a median of 3,288 (IQR: 1,298

−12,620). The average ages of patients ranged from 49.3 ± 15.3 to 71.6 ± 13.2 years. Female patients represented 30–60% of patients in the included studies. Three studies reported characteristics and changes in ED LOS specific to discharged patients, while the remaining studies reported characteristics and changes in ED LOS for all included/enrolled patients.

## Methodological quality of the included studies

The overall risk of bias for the five included randomized trials was low (S1 Fig). Three RCTs were open-label although otherwise methodologically rigorous. The included observational studies were classified as having intermediate or low risk of bias (S2 Fig), with a median NOS score (8; IQR, 6–8) representing an overall low risk of bias.

## Primary outcome: Impact of ADP implementation on ED LOS

A significant reduction in ED LOS was reported in 22 observational studies and four RCTs (Table 2). Six studies showed statistically non-significant changes in ED LOS after ADP implementation, whereas five studies reported a significant increase. In the simple linear meta-regression, each one hour increase in the pre-interventional ED LOS was associated with a 30-minute (95% CI, 21.6–38.4; $p$-value <0.001) reduction in ED LOS following ADP implementation (Fig 2). Among the seven studies with a baseline ED LOS of ≤4 hours, two showed either no change or an increase in LOS, while the remaining four reported reductions of <25 minutes.

Eight studies did not explicitly state a serial Tn time used in the pre-intervention phase. These studies were excluded from any meta-analysis examining the effect of a change in the Tn measurement interval. A pre-planned meta-analysis stratified by the change in the serial Tn measurement interval was conducted. The overall statistical heterogeneity was high, and the pooled results should be interpreted cautiously. Considering subgroups, studies that reported a delta change of <1 hour were more homogenous with a pooled median reduction of 25.2 minutes (-0.42 h; 95% CI, -0.49 to -0.35) (Fig 3). The remaining subgroups demonstrated significant heterogeneity ($I^2$ >99%). The median of median differences was consistent with the pooled estimate for each subgroup with 24-, 30-, and 63-minute reductions for studies that reported 0–1-hour, 2–3-hour, and >3-hour reductions in Tn interval, respectively.

Eighteen studies transitioned from a conventional to hsTn. Eleven of these studies demonstrated a significant decrease in ED LOS following ADP implementation whereas three showed a significant increase [21, 30, 45]. When pooled for meta-analysis, there was high heterogeneity among these studies ($I^2$ >99%) and the results should be interpreted cautiously; however, transitioning from conventional Tn to hsTn impacted median ED LOS (-0.50 hours; IQR: -0.88 to -0.11) (Fig 4).

A random-effects linear meta-regression was performed to determine which clinical parameters had the largest influence on the change in ED LOS. Specifically, we were interested in changes in serial Tn measurement intervals, transition from conventional Tn to hsTn assays, and pre-intervention ED LOS. Neither change in serial Tn measurement interval nor transition from conventional Tn to hsTn assays were found to significantly impact a change in ED LOS in the univariate analyses. Pre-interventional ED LOS, however, was significantly associated with a reduction in ED LOS in the linear meta-regression after adjusting for both the change in serial Tn measurement interval and transition from conventional Tn assay to an hsTn assay (β = -0.26; 95% CI, -0.43 to -0.10; $p$-value = 0.001).

## Secondary outcome: Impact of ADP implementation on admissions

Twenty-four studies reported admission proportions (S1 Table). Given the wide variety of countries and healthcare regions involved, there was a correspondingly large range of

**Table 2. Summary of troponin types used, delta troponin measurement intervals, and ED LOS reported before and after ADP implementation among the included studies.**

| Author | Sample size | | Tn used | | Tn interval (hours) | | ADP used | ED LOS reported, h; median (IQR)/mean ± SD/ difference (IQR) | | | p-value |
|---|---|---|---|---|---|---|---|---|---|---|---|
| | Pre | Post | Pre | Post | Pre | Post | | Pre | Post | Difference | |
| **Before-after studies** | | | | | | | | | | | |
| Al Marashi 2020 | 308 | 326 | hsTnI | hsTnI | NS | 3 | NS | 8.60 | 5.20 | | 0.001 |
| Allen 2018 | 15946 | 15144 | NS | NS | 6 | 3 | HEART | 6.48 ± 0.29 | 6.62 ± 0.51 | | 0.380 |
| Barnes 2021 | 1131 | 1124 | hsTnI | hsTnI | 3 | 2 | HEART | 4.30 (3.30−7.10) | 3.60 (2.60−5.40) | | 0.001 |
| Bevins 2022 | 3641 | 4203 | hsTnT | hsTnT | NS | 1 | ESC algorithm | 8.10 | 7.08 | | <0.001 |
| Buttinger 2019 | 391 | 491 | hsTnT | hsTnT | 3 | 1 | ESC algorithm | 3.92 (3.18−6.23) | 3.78 (3.20−5.17) | | |
| Crowder 2015 | 6866 | 5754 | cTnT | hsTnT | 6 | 2 | NS | 6.60 (4.25−9.80) | 6.10 (4.12−8.73) | | 0.001 |
| Ford 2021 | 1589 | 1616 | cTnI | hsTnT | 3 | 1 | HEART | 6.20 (4.20−9.40) | 6.40 (4.30−9.60) | | |
| Furmaga 2021 | 4892 | 7453 | cTn | hsTnT | NS | NS | HEART | 3.47 (2.45−4.73) | 3.83 (2.67−5.25) | | 0.010 |
| Ganguli 2021 | 3665 | 3899 | cTn | hscTn | NS | 1 | NS | 7.44 ± 8.16 | 6.24 ± 6.96 | | |
| Greenslade 2020 | 5764 | 6866 | cTn | TnI | 6 | 2 | IMPACT | 9.00 (5.9−14.8) | 7.40 (4.80−12.10) | | |
| Hill 2023 | 1333 | 1307 | cTnI | hsTnI | 3 | 3 | HEART | 6.53 | 6.18 | -0.35 (95% CI, -0.61 to -0.09) | |
| Hughes 2023 | 31875 | 27357 | cTnI | hsTnI | 4 | 2 | NS | 5.90 | 5.80 | -0.10 (95% CI, -0.20 to 0.00) | |
| Ljung 2019 | 612 | 621 | cTn | hsTn | 3 | 1 | HEART | 3.80 (3.10−4.90) | 4.00 (2.40−4.80) | | |
| Mahler 2018 | 3713 | 4761 | hsTnI | hsTnI | 3 | 3 | HEART | 4.00 (2.80−5.20) | 3.60 (2.60−5.00) | | 0.150 |
| Mohmed 2021 | 1642 | 1376 | NS | hsTnT | NS | 3 | ESC 0/3-h ADP | 16.30 | 7.10 | | 0.001 |
| Mountain 2016 | 426 | 603 | hsTnI | hsTnI | 8 | 4 | NS | 6.90 | 5.50 | | 0.001 |
| Mumma 2020 | 540 | 538 | cTnI | hsTnT | NS | 1 | HEART | 7.10 (5.30−10.60) | 8.10 (5.20−11.50) | | |
| Mungai 2020 | 150 | 150 | cTnT | hsTnT | 3 | 1 | NS | 8.40 (1.90−14.80) | 3.90 (0.50−8.50) | | 0.001 |
| Ola 2021 | 1738 | 1798 | cTnT | hsTnT | 3 | 2 | NS | 4.30 (2.30−46.10) | 4.20 (2.70−47.90) | | 0.010 |
| Parsonage 2017 | 30769 | 23699 | cTn | hsTnI | 8 | 2 | ADAPT | 3.80 (2.70−5.80) | 3.50 (2.40−4.90) | | 0.010 |
| Phillips 2023 | 4905 | 6798 | hsTnT | hsTnT | NS | 2 | NS | 9.00 (2.90−24.20) | 8.20 (2.80−21.40) | | 0.050 |
| Randolph 2018 | 4295 | 769 | hsTn | hsTnT | 6 | 2 | NS | 3.30 (3.20−3.50) | 3.30 (3.00−3.70) | | 0.960 |
| Rowe 2023 | 2031 | 2308 | cTn | hsTnI | 6 | 3 | NS | 7.17 | 6.67 | -0.50 (95% CI, -0.80 to -0.21) | |
| Ruangsomboon 2018 | 65 | 65 | hsTnT | hsTnT | 3 | 1 | NS | 4.30 (3.00−5.40) | 2.30 (1.80−3.70) | | 0.001 |
| Suh 2022 | 1071 | 821 | cTnT | hsTnT | 3 | 1 | ESC 0/1-h ADP | 11.30 (8.10 −20.20) | 11.50 (7.60 −22.90) | | 0.962 |
| Than 2018 | 11529 | 19803 | NS | variable | 6 | 2 | Multiple | | | -2.9 (-3.4 to -2.4) | 0.001 |
| Than 2021 | 1073 | 1343 | hsTnI | hsTnI | 2 | 2 | COVID Path | 3.80 (2.80−4.90) | 3.40 (2.60−4.60) | | <0.001 |
| Trent 2022 | 649 | 649 | hsTn | hsTn | 3 | 1 | Other | 4.70 | 3.20 | -1.13 (-1.50 to -0.77) | |
| Twerenbold 2016 | 1455 | 1089 | cTnT | hsTnT | 6 | 3 | NS | 6.30 (3.80−8.70) | 5.10 (3.60−7.20) | | 0.046 |
| Tyner 2023 | 7117 | 7623 | hsTn | hsTnI | 6 | 3 | HEART | 5.20 (2.90−8.20) | 4.40 (2.70−5.90) | | 0.036 |
| VanAssche 2023 | 100 | 100 | cTnI | hsTnI | 3 | 1 | NS | 5.27 | 4.82 | | 0.090 |
| Vigen 2020 | 16991 | 14552 | cTn | hsTnT | NS | 1 | HEART | 6.42 (4.67−9.68) | 6.52 (4.87−9.27) | | |
| **RCTs** | | | | | | | | | | | |
| | Ctrl. | Exp. | Ctrl. | Exp. | Ctrl. | Exp. | | Ctrl. | Exp. | | |
| Anand 2021 | 14700 | 16792 | hsTnI | hsTnI | 6 | 3 | Other (STEACS) | 10.10 ± 4.10 | 6.80 ± 4.10 | | 0.001 |
| Chew 2019 | 1642 | 1646 | hsTnT | hsTnT | 3 | 1 | Multiple | 5.60 (4.00−7.10) | 4.60 (3.40−6.40) | | 0.001 |
| Carlton 2020 | 313 | 316 | hsTn | hsTn | 1 | 0 | NS | 5.00 (3.40−7.40) | 4.40 (3.20−6.80) | | |
| Lambrakis 2021 | 1632 | 1638 | cTn | hsTnT | 3 | 1 | NS | 6.30 (4.80−18.30) | 4.60 (3.50−7.60) | | |

(*Continued*)

**Table 2.** (Continued)

| Author | Sample size | | Tn used | | Tn interval (hours) | | ADP used | ED LOS reported, h; median (IQR)/mean ± SD/ difference (IQR) | | | *p*-value |
|---|---|---|---|---|---|---|---|---|---|---|---|
| | Pre | Post | Pre | Post | Pre | Post | | Pre | Post | Difference | |
| Miller 2022 | 13505 | 19104 | NS | hsTnI | 3 | 1 | NS | | | +0.77 h longer than standard care | |

ADP, accelerated diagnostic protocol; cTn, conventional troponin; Ctrl., control group; ED, emergency department; ESC, European Society of Cardiology; Exp., experimental group; HEART, history, ECG, age, risk factors, and troponin; hsTn, high-sensitivity troponin; STEACS, High-Sensitivity Troponin in the Evaluation of patients with suspected Acute Coronary Syndrome; IMPACT, Improved Assessment Of Chest Pain Trial; IQR, interquartile range; LOS, length of stay; NS, not stated; RCT, randomized controlled trial; SD, standard deviation; Tn, troponin; Pre, before ADP implementation; Post, after ADP implementation.

admission proportions from 8.8% [19] to 68.3% [33]. Three studies reported an increase (range, +0.9 to +5.0%) in the proportion of patients admitted after introducing an ADP [25, 30, 34], one of which was significant [25]. Fifteen studies reported a significant decrease (range, −1.2 to −26.0%) in admission proportion with changes observed mostly within the first 24 to 48 hours post-presentation. There was high heterogeneity among the 24 studies that reported admission proportions, but with a statistically significant pooled estimate of RR = 0.84 (95% CI, 0.79−0.89) (S3 Fig).

### Secondary outcome: Impact of ADP implementation on MACE incidence

MACE occurrence within 30 days was reported in 10 observational studies and three RCTs [46−48]. One observational study [29] reported a 45-day MACE. Pre-interventional MACE incidence ranged from 0.4−19.0%, while post-interventional rates ranged from 0.3−14.9%. Five studies reported increases in MACE incidence (range, +0.1 to +3.0%), none of which was significant. On the other hand, six studies reported reductions in MACE occurrence (range, −0.1 to −6.2%), only one of which was a significant reduction. There was moderate heterogeneity among the 13 studies that reported MACE proportions, with a statistically non-significant pooled estimate of RR = 0.95 (95% CI, 0.86−1.04) (S4 Fig).

### Discussion

This systematic review provides a comprehensive summary of the operational and patient-oriented clinical outcomes following the implementation of ADPs in ED settings for suspected cardiac chest pain. Overall, 37 studies were identified for inclusion in this review. While international settings were represented and demonstrated interest in the topic, most of the published research involved North American EDs. Likewise, the included studies were diverse in size, ranging from small observational studies to large RCTs or studies utilizing administrative datasets. Notably, despite the limitations of data pooling due to heterogeneity among the studies, our results show that using an ADP, generally accompanied by a hsTn, resulted in lower ED LOS in a majority of studies. Overall, 68% of studies demonstrated a significant reduction in ED LOS. The pooled effect for all included studies was a reduction of just over 1 hour (Fig 3); in the context of ED crowding, a change of 1 hour represents a clinically significant improvement.

These findings are important because while clinical gestalt remains a critical component of decision-making in patients with typical clinical presentations and high troponin levels, ADPs complement clinical gestalt by providing a reliable, evidence-based framework for managing patients with chest pain. This framework mitigates the variability associated with gestalt alone, ensuring that most patients are discharged safely and efficiently. The overall results provide

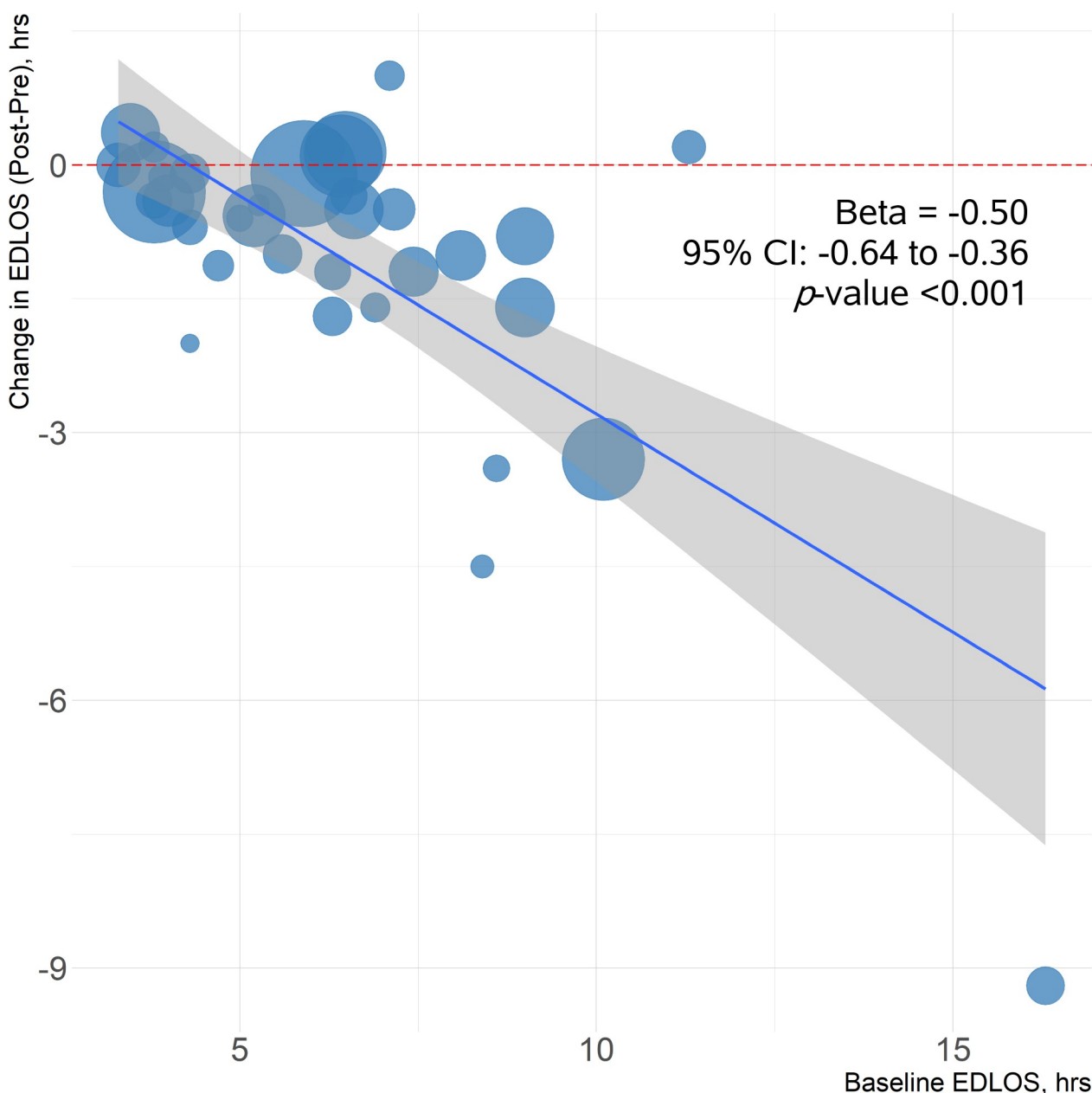

**Fig 2. Bubble plot illustrating the association between reduction in length of stay and pre-interventional emergency department length of stay.** The size of each bubble is proportional to the sample size of the study it represents.

direction for clinicians and administrators and the additional findings below provide opportunities for further refinement and implementation research.

There was a significant association between pre-interventional ED LOS and post-intervention reduction in ED LOS (Fig 2). This relationship was again reflected in the meta-regression after adjusting for serial Tn measurement intervals and transition from a conventional Tn assay to an hsTn assay. It is reasonable to expect a larger change in LOS when initial ED wait times are longer. One consideration for the minimal reductions in LOS seen in these EDs may be due to wait times rather than protocol efficiency. Patient wait times (e.g., time from triage

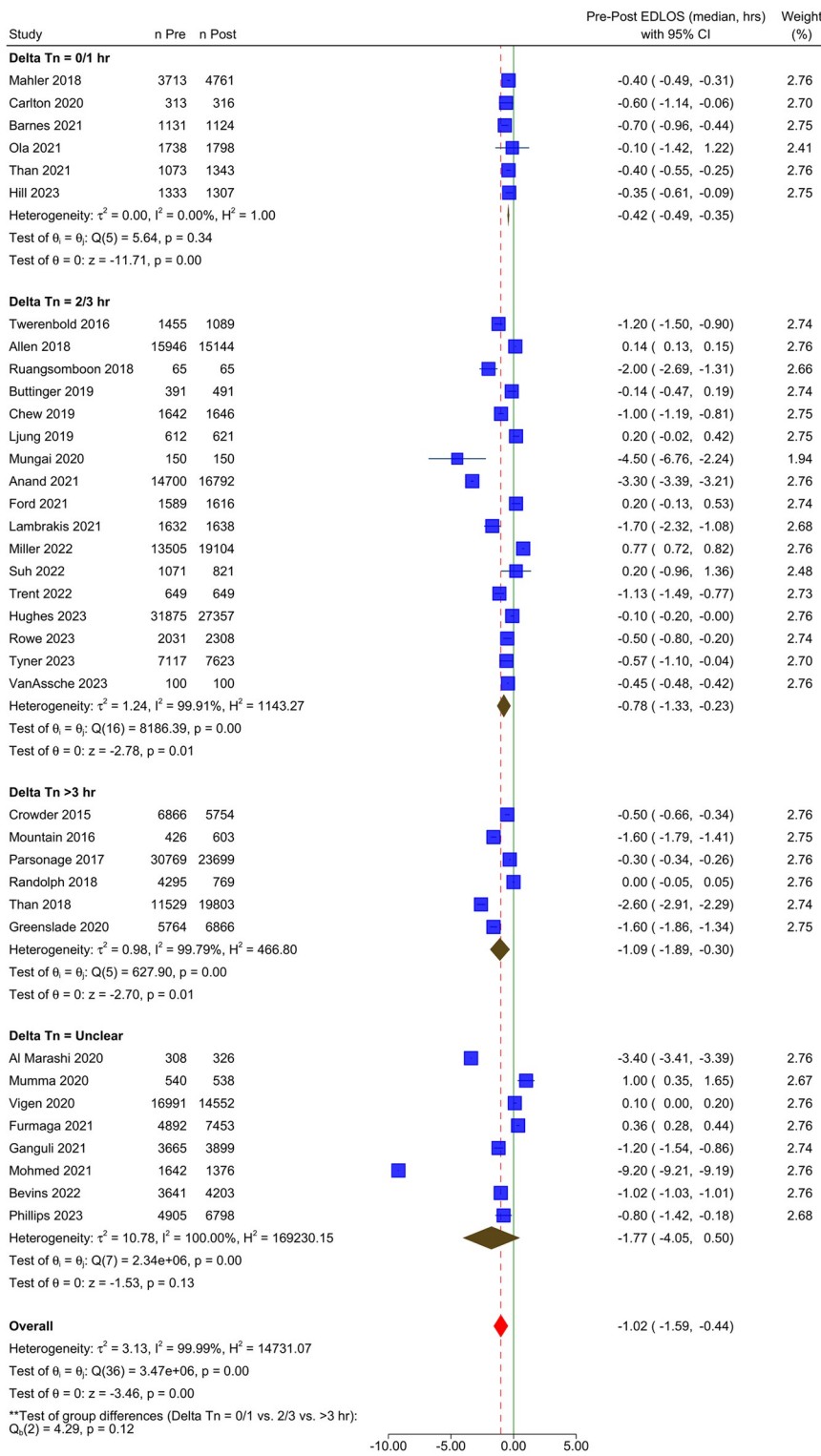

**Fig 3. Forest plot showing the effect of delta troponin measurement interval timing on emergency department length of stay.**

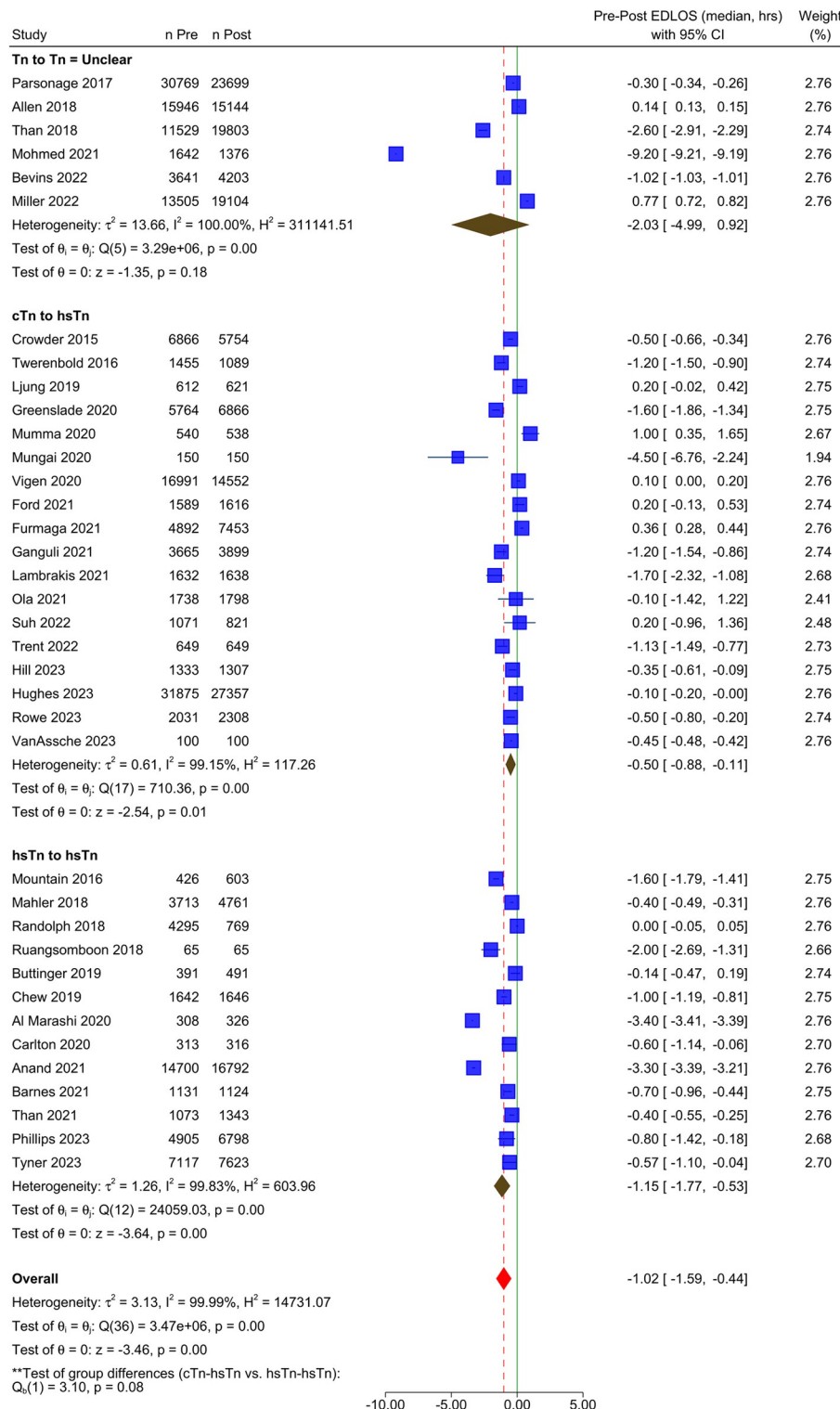

**Fig 4. Forest plot showing the effect of transitioning from conventional to high-sensitivity troponin assay vs. using a high-sensitivity troponin assay in the pre- and post-implementation periods on emergency department length of stay.**

to room placement and time from placement to physician initial assessment [PIA]) were not frequently reported in the included studies, and while there is certainly considerable variation among healthcare systems, many patients are likely waiting up to several hours prior to PIA. Given that this time prior to physician assessment is not modifiable by an ADP, sites with a low baseline ED LOS may have very little modifiable time and hence their corresponding modest reductions.

Another possible explanation is that a significant number of patients undergo only a single Tn measurement. If a patient is evaluated in a time frame after the serial measurement interval dictated by whichever Tn protocol is being employed, then a single initial negative result may be sufficient to rule out cardiac ischemia. In this scenario, the addition of an ADP would be expected to have little to no impact, as patients dispositioned after a single Tn assay are unaffected by the timing of serial measurement. Most studies did not restrict enrollment to patients undergoing two or more Tn measurements, nor did they report the results for the subgroup. In the two studies that did restrict enrollment, reductions in ED LOS of over 1.5 hours were observed. We recommend that future publications aim to include more robust reporting for ED LOS outcomes, more granular reporting of ED times such as PIA, and stratify patients based on the number of Tn tests ordered. This will improve the evaluation of the effectiveness of interventions and provide an assessment of whether these protocols truly help improve efficiency.

Several authors cited a lack of familiarity with and uptake of ADP usage [14, 21, 27, 42] as possible factors contributing to the modest reductions in ED LOS. In other studies, this was anticipated and mitigated. The trial design for the study conducted by Anand et al. [46] was particularly robust and included a randomization period of several months where hospitals and practitioners became familiar with the usage of the ADP and an assay, prior to the true implementation phase. In addition to eliminating very lengthy serial Tn measurement intervals (up to 12 hours), this familiarization period likely contributed to the large effect size observed. Both studies that showed a significant increase in ED LOS implemented the HEART score in conjunction with a new hsTn assay [21, 30]. While the authors did not state any specific Tn measurement interval prior to their studies, Furmaga et al. [21] suggested that the new measures encouraged more frequent serial testing as well as some provider unfamiliarity with using an ADP which drove the increase in time spent in the ED.

The management of ED presentations for chest pain also varied among some of the regions with included studies. This is perhaps best illustrated by the difference between the admission proportions in Canada compared to Australia. Among the Canadian studies, only 8.8–25.1% of patients were admitted and there was a strong focus on outpatient management and follow-up; however, in the Australian studies reporting admissions data, the proportion ranged from 24.0–68.3%. This variation was likely due to the widespread use of observational units for the further assessment of chest pain in Australia. One UK study [28] was noted to be a significant outlier with pre-intervention ED LOS in excess of 16 hours. The implementation of an ADP reduced ED LOS to 7.1 hours which is more in keeping with the other studies. This reduction was largely driven by a major reduction in the proportion of patients being admitted (presumably to an observation unit, although not specified in the abstract), a decrease from 73% to 44%. A subgroup analysis of only discharged patients revealed a more modest 48-minute reduction in ED LOS between treatment groups. Despite the differences in practice, it is reassuring that only 3 studies [25, 30, 34] reported an increase in admissions following ADP introduction; only one of these reported a significant increase [25].

One of the earlier concerns regarding the implementation of ADP was the possibility that a higher proportion of patients would experience adverse outcomes following rapid discharge due to decreased observational time in the ED. In this review, using MACE as an important

30-day outcome, no such increase in adverse events was identified. Two large studies reported the lowest proportion of patients experiencing MACE at 0.30% and 0.35% [14, 46]. On the high end, a MACE proportion of up to 19% was reported in one study [37]. Another study [15] showed a significant decrease in the MACE proportion after implementing an ADP (RR, 0.56; 95% CI, 0.40–0.79). None of the studies included in the present review reported a significant increase in MACE after implementing an ADP, suggesting that the implementation of chest pain protocols is safe.

## Strengths and limitations

This review has both strengths and limitations that require discussion. A comprehensive search strategy was created with the help of a health sciences librarian and generated a robust number of studies for review. Efforts aimed at mitigating publication bias were employed; however, we recognize that some publications could have been missed. Our review was further strengthened by protocol registration and efforts to avoid selection bias. In addition, the data span numerous countries and healthcare regions which increases the external validity of our results.

Our results are limited by the fact that the included studies are predominantly observational in design. Many of the included ED studies reported LOS as a secondary outcome and as such variable outcome reporting was common. There was also considerable variation in the types of ADP employed as well as the type and timing of serial Tn measurements. The inclusion of studies utilizing different ADPs introduces additional heterogeneity due to the variability in the criteria used. While most ADPs share common components, differences such as the inclusion of syncope and CK-MB in the IMPACT protocol may influence outcomes. Outcome measures such as ED LOS were somewhat obscured by reporting results for all patients who received a Tn assay. Granular reporting of LOS results for the subset of patients who underwent serial Tn measurements would allow for more definitive conclusions. There was a large degree of statistical heterogeneity across most subgroups and RCT data were combined with observational data; thus, the results of the pooled analyses should be interpreted cautiously. In particular, the definition of ED LOS varied among the included studies (S2 Table), with a large majority defining it as the time from ED arrival to either discharge or disposition. Two studies did not use either definition; Suh et al. [38] defined ED LOS as the time from first clinician provider evaluation to disposition decision time (provider-to-disposition time), while Anand et al. [46] defined it as the time from ED arrival (presentation) to hospital discharge. Again, a number of included studies did not specify the ADP used. A sensitivity analysis comparing studies that reported versus did not report the ADP type, however, showed very similar and overlapping effect estimates, indicating that the absence of specific ADP information did not substantially impact the overall findings. We concluded that with similarities in samples, presenting complaints, and intervention implementation, pooling was valid. While this variability speaks to the broad applicability of ADP effectiveness, it does obscure the comparative effectiveness of individual elements of the ADP.

We were also concerned about the skewed data associated with ED LOS reporting, as there are often outliers with dramatically higher LOS values. From the available data, we estimated means and standard deviations using the method proposed by Wan et al. [51] to assess for differences and found no major changes from the original median and IQR data pooling. We, therefore, believed it was most appropriate to report the data as the original medians and IQRs.

## Conclusions

This systematic review examined the operational and clinical outcomes of ADPs implemented for patients presenting to EDs with suspected cardiac chest pain. Our findings showed that

implementing an ADP may significantly reduce ED LOS and should be considered by hospitals or healthcare entities searching for strategies to improve operational efficiency. This decrease in LOS was even seen in the absence of any change in the type of Tn assay. Moreover, this decrease in LOS was associated with meaningful reductions in hospital admissions, without an increase in subsequent adverse events (such as MACE). The observed benefits also translated across multiple countries and health regions. Further research should evaluate the optimal combination of Tn measurement interval in combination with specific ADPs.

## Supporting information

**S1 Fig. Risk of bias assessments for controlled trials.**
(TIF)

**S2 Fig. Risk of bias assessments for observational studies.**
(TIF)

**S3 Fig. Impact of ADP implementation on the proportions of chest pain patients admitted.**
(TIF)

**S4 Fig. Impact of ADP implementation on the incidence of major adverse cardiovascular events.**
(TIF)

**S1 Table. Proportions of admitted patients and those experiencing MACE (major adverse cardiac events) within 30 days after ADP implementation.**
(DOCX)

**S2 Table. Definitions of ED LOS used among the included studies.**
(DOCX)

**S1 File. Search strategy used in the systematic review.**
(DOCX)

**S2 File. List of included studies (ordered by study ID).**
(DOCX)

**S3 File. Characteristics of excluded studies (ordered by study ID).**
(DOCX)

**S4 File. Data extracted from the included studies.**
(XLSX)

**S5 File. Risk of bias domains and overall ratings for the included observational studies (Newcastle−Ottawa scores).**
(DOCX)

**S6 File. Risk of bias domains and overall ratings for the included randomized controlled studies (assessed using the Cochrane Risk of Bias tool).**
(DOCX)

**S7 File. PRISMA 2009 checklist.**
(DOC)

**S8 File. PROSPERO registration CRD42021249679.**
(PDF)

## Author Contributions

**Conceptualization:** Jesse Hill, Brian H. Rowe.

**Data curation:** Jesse Hill, Nana Owusu M. Essel, Esther H. Yang, Liz Dennett, Brian H. Rowe.

**Formal analysis:** Jesse Hill, Nana Owusu M. Essel, Esther H. Yang, Brian H. Rowe.

**Funding acquisition:** Brian H. Rowe.

**Investigation:** Jesse Hill, Esther H. Yang, Liz Dennett, Brian H. Rowe.

**Methodology:** Nana Owusu M. Essel, Esther H. Yang, Brian H. Rowe.

**Project administration:** Jesse Hill, Brian H. Rowe.

**Resources:** Esther H. Yang, Liz Dennett, Brian H. Rowe.

**Software:** Nana Owusu M. Essel.

**Supervision:** Brian H. Rowe.

**Validation:** Jesse Hill, Nana Owusu M. Essel, Esther H. Yang, Brian H. Rowe.

**Visualization:** Nana Owusu M. Essel.

**Writing – original draft:** Jesse Hill, Nana Owusu M. Essel.

**Writing – review & editing:** Jesse Hill, Nana Owusu M. Essel, Esther H. Yang, Liz Dennett, Brian H. Rowe.

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
