## [Decision Letter · Decision Letter 0]

18 Jun 2024

PONE-D-24-12460Effectiveness of accelerated diagnostic protocols for reducing emergency department length of stay in patients presenting with chest pain: A systematic reviewPLOS ONE

Dear Dr. Essel,

Thank you for submitting your manuscript to PLOS ONE. After careful consideration, we feel that it has merit but does not fully meet PLOS ONE’s publication criteria as it currently stands. Therefore, we invite you to submit a revised version of the manuscript that addresses the points raised during the review process.

We look forward to receiving your revised manuscript.

Kind regards,

Daniel Antwi-Amoabeng, MD, MSc

Academic Editor

PLOS ONE

Journal Requirements:

2. Please identify your study as "systematic review and meta-analysis" in the title of your manuscript.

Dr. Jesse Hill has no funding sources to report. Dr. Brian Rowe’s research is supported by a Scientific Director’s Grant (SOP 168483) from the Canadian Institutes of Health Research (CIHR, Ottawa, Ontario). Ms. Esther Yang is supported by the Emergency Medicine Research Group (EMeRG) in the Department of Emergency Medicine, University of Alberta. The funders take no responsibility for the conduct, analyses, and interpretation of these results.

Additional Editor Comments:

This work reports the utility of accelerated diagnostic protocols for chest pain in improving operational efficiency. While the primary outcome is not patient-centered, and thus, limit the value of the conclusions drawn to daily practice, documenting the role of APDs in process improvement is a worthwhile endeavor and I commend the authors for undertaking this task masterfully.

Abstract

Well written.

Rationale clearly stated.

Line 41: Please state the follow-up time for the MACE assessment here.

Lines 44 – 47: Is difficult to understand. Please consider re-writing for clarity.

Introduction

Well written.

Methods

Admission proportion: Is there a baseline admission rate?

Results:

Table 1: please define what the “Before”, “After” columns mean.

Lines 200 – 201: Which changes are being referred to here? ED LOS?

Table 2: edit the title to include reference to before and after ADP implementation.

Table 2: what is the unit of the Tn interval measure?

Discussion:

Line 288: LOS probably belongs somewhere in the sentence.

Given the large heterogeneity in the studies, perhaps a thematic review of the observational studies to tease out the best parts of each observational study may have been more beneficial instead of a meta-analysis.

Line 359 – 360: If the claim that pooling was valid based on the similarities in the patient population, could the heterogeneity be at the level of which ADP was implemented? Sub-group analysis by ADP type would likely have more clinical value.

So, it is the specific algorithm not troponin assay type, which seems to drive the reduced LOS. Is this stamen correct?

Please comment on the performance of ADPs in general against clinical gestalt in terms of ED LOS. In my experience, a convincing story with a high Tn usually leads to an admission. So, the value of the ADP perhaps lies in those scenarios where the story is not clear. The clinical utility of this study may have been enhanced if the authors focused on those kinds of ADPs which addressed this scenario.

Conclusion:

The first sentence was not a stated objective of the study. Please edit.

Line 372: Please edit this sentence. Since observational studies were included in the LOS estimates, caution should be exercised when drawing causative conclusions.

Reviewers' comments:

Reviewer's Responses to Questions

**Comments to the Author**

1. Is the manuscript technically sound, and do the data support the conclusions?

Reviewer #1: Partly

Reviewer #2: Yes

2. Has the statistical analysis been performed appropriately and rigorously? 

Reviewer #1: Yes

Reviewer #2: Yes

3. Have the authors made all data underlying the findings in their manuscript fully available?

Reviewer #1: Yes

Reviewer #2: Yes

4. Is the manuscript presented in an intelligible fashion and written in standard English?

Reviewer #1: Yes

Reviewer #2: Yes

5. Review Comments to the Author

**Reviewer #1: **Thank you for the opportunity of reviewing the manuscript entitled “Effectiveness of accelerated diagnostic protocols (ADPs) for reducing emergency department length of stay in patients presenting with chest pain: A systematic review”. The authors performed a novel systematic review on the usefulness of ADPs on the main outcome of length of stay (LOS) in the emergency department (ED). Meta-analysis was done but the authors genuinely did not reflect it in the title due to the high rate of heterogeneity. They fairly considered potential confounders and have them adjusted in meta-regression.

Previous systematic reviews and meta-analyses have addressed the efficacy of HEART score for detecting MACE, which was a secondary outcome in this study. How this study will affect clinical practice? Nowadays, most clinicians use ADPs. The results regarding hsTn vs. Tn in terms of LOS seems to be more practical.

There are some comments regarding this article:

Major points:

Abstract:

• It has been concluded that ADPs can reduce LOS but did not change admission rates although in the result section it is mentioned that ADPs significantly changed the admission rate. This needs to be clarified.

Methods:

• In this systematic review, not a single test (criteria) but studies with various ADPs were included such as HEART pathway, IMPACT, COVID path, ESC algorithm, ADAPT, STEACS and their data were pooled. Although most of the ADPs consist of same items, still there are some different items for example: syncope and CK-MB in IMPACT. This can result in a high risk of bias, which should be addressed and mentioned in the limitation section.

• Moreover, 16/37 studies, including 3/5 RCTs, did not state which ADP they used. The meta-analysis could be done excluding these studies regardless of the overall ROB score.

• The definition for LOS was different among various studies: for instance,

Ref 21: the definition for the ED LOS: from patients’ arrival to ED disposition

Ref 45: from patients’ arrival to ED departure

They should cautiously pool these data and mention this issue in the limitation section.

Results:

• For secondary objectives, clinically practical data of admission days and MACE can be elaborated. For example, what was the time range for changes in admission rate?

• I found the S1 supplementary table very useful as well even though they were secondary outcomes, they present practical outcomes for clinicians.

Conclusions:

• The authors fairly mentioned the limitations of this study and accordingly, the conclusion should be revised and interpreted cautiously since the heterogeneity was high (>99% for the primary outcome). It is recommended to mainly focus on descriptive reports of the study results and not to optimistically rely on meta-analysis.

Minor Point:

• This sentence need to be clarified about “changes”, Line 200-201:

“Three studies reported changes specific to discharged patients, while the remaining studies reported changes for all patients.” For example, were discharged patients younger or without past medical history, etc? What did authors mean by specific changes for all the patients?

**Reviewer #2: **The article titled "Effectiveness of Accelerated Diagnostic Protocols for Reducing Emergency Department Length of Stay in Patients Presenting with Chest Pain: A Systematic Review" offers a comprehensive analysis of the impact of accelerated diagnostic protocols (ADPs) on the efficiency and operational metrics for evaluating suspected cardiac chest pain in emergency departments.

As noted by the authors, previous studies have established the safety of accelerated chest pain protocols, showing similar proportions of admissions and Major Adverse Cardiac Events (MACE) compared to traditional methods. In this study, the authors investigate how effectively ADPs improve patient flow through emergency departments by analyzing their impact on operational metrics. The review encompasses 37 articles involving over 404,566 patients across various countries and healthcare settings, enhancing the generalizability of the findings. The study adhered to PRISMA guidelines and was registered in PROSPERO. Risk of bias was assessed using established tools, and data extraction was performed independently by multiple reviewers.

While it is intuitive to expect these protocols to improve patient flow, the authors' key findings are significant: ED Length of Stay (LOS) showed substantial reductions, especially in departments with higher baseline LOS; implementing ADPs did not lead to an increase in adverse cardiac events, thereby maintaining patient safety; and there was a decrease in admission rates after implementation, indicating a reduction in unnecessary admissions without compromising patient safety. The authors also address and discuss the limitations of their work, including heterogeneity in study design and reporting variability.

This study is a valuable contribution to emergency medicine, and I recommend acceptance without further revisions.

6. PLOS authors have the option to publish the peer review history of their article (what does this mean?). If published, this will include your full peer review and any attached files.

Reviewer #1: No

Reviewer #2: **Yes: **Dr. Daniel Robert Beamish

---

## [Author Response · Author response to Decision Letter 0]

23 Jul 2024

July 23, 2024

Daniel Antwi−Amoabeng, MD, MSc

Academic Editor

PLOS ONE

Dear Dr. Antwi−Amoabeng,

RE: Manuscript ID PONE-D-24-12460.

Thank you for your revise and resubmit decision on our article. We are pleased to re-submit the manuscript with the revised title “Effectiveness of accelerated diagnostic protocols for reducing emergency department length of stay in patients presenting with chest pain: A systematic review and meta-analysis.” 

We thank the PLOS ONE editorial team and the reviewers for the thoughtful and detailed comments and recommendations on the manuscript. We have revised it to address these concerns as requested. We feel the thoughtful and probing suggestions have greatly improved the manuscript. The responses to all comments have been prepared and attached.

As per the requirement, we wish to update our funding statement as follows: 

Dr. Jesse Hill has no funding sources to report. Dr. Brian Rowe’s research is supported by a Scientific Director’s Grant (SOP 168483) from the Canadian Institutes of Health Research (CIHR, Ottawa, Ontario). Ms. Esther Yang is supported by the Emergency Medicine Research Group (EMeRG) in the Department of Emergency Medicine, University of Alberta. There was no additional external funding received for this study. The funders take no responsibility for the conduct, analyses, and interpretation of these results.

We look forward to your response to our revisions. If there is anything else required, please do not hesitate to contact me.

Yours sincerely,

Brian H. Rowe, MD, MSc, CCFP(EM), FCFP, FCCP, FCAHS

Professor, Department of Emergency Medicine, Faculty of Medicine and Dentistry

Professor, School of Public Health

College of Health Sciences, University of Alberta

Edmonton, Alberta

---

## [Decision Letter · Decision Letter 1]

12 Aug 2024

Effectiveness of accelerated diagnostic protocols for reducing emergency department length of stay in patients presenting with chest pain: A systematic review and meta-analysis

PONE-D-24-12460R1

Dear Dr. Essel,

We’re pleased to inform you that your manuscript has been judged scientifically suitable for publication and will be formally accepted for publication once it meets all outstanding technical requirements.

Kind regards,

Martin E. Matsumura, MD

Academic Editor

PLOS ONE

Additional Editor Comments (optional):

All reviewer comments have been adequately addressed by the authors

Reviewers' comments:

Reviewer's Responses to Questions

**Comments to the Author**

1. If the authors have adequately addressed your comments raised in a previous round of review and you feel that this manuscript is now acceptable for publication, you may indicate that here to bypass the “Comments to the Author” section, enter your conflict of interest statement in the “Confidential to Editor” section, and submit your "Accept" recommendation.

Reviewer #2: All comments have been addressed

2. Is the manuscript technically sound, and do the data support the conclusions?

Reviewer #2: Yes

3. Has the statistical analysis been performed appropriately and rigorously? 

Reviewer #2: Yes

4. Have the authors made all data underlying the findings in their manuscript fully available?

Reviewer #2: Yes

5. Is the manuscript presented in an intelligible fashion and written in standard English?

Reviewer #2: Yes

6. Review Comments to the Author

Reviewer #2: The authors have substantially addressed the requested minor revisions. In my opinion this article should be accepted for publication without further revision.

7. PLOS authors have the option to publish the peer review history of their article (what does this mean?). If published, this will include your full peer review and any attached files.

Reviewer #2: **Yes: **Dr. Daniel Robert Beamish MD, MA, PhD, RMSK, CCFP(EM)

---

## [Editor Report · Acceptance letter]

21 Aug 2024

PONE-D-24-12460R1 

PLOS ONE

Dear Dr. Essel, 

I'm pleased to inform you that your manuscript has been deemed suitable for publication in PLOS ONE. Congratulations! Your manuscript is now being handed over to our production team.

Kind regards, 

on behalf of

Dr. Martin E. Matsumura 

Academic Editor

PLOS ONE